# Current State of In Vitro Embryo Production in African Lion (*Panthera leo*)

**DOI:** 10.3390/ani12111424

**Published:** 2022-05-31

**Authors:** Jennifer Zahmel, Kim Skalborg Simonsen, Julia Stagegaard, Sergio Eliseo Palma-Vera, Katarina Jewgenow

**Affiliations:** 1Department of Reproduction Biology, Leibniz-Institute for Zoo and Wildlife Research, Alfred-Kowalke-Str. 17, 10315 Berlin, Germany; palma-vera@izw-berlin.de (S.E.P.-V.); jewgenow@izw-berlin.de (K.J.); 2Givskud Zoo—Zootopia, Løveparkvej 3, 7323 Give, Denmark; kss@givskudzoo.dk; 3Ree Park—Safari, Stubbe Søvej 15, 8400 Ebeltoft, Denmark; julia.stagegaard@reepark.dk

**Keywords:** assisted reproduction, in vitro fertilization, in vitro maturation, African lion, domestic cat

## Abstract

**Simple Summary:**

Many cat species reproduce less successfully in human care than in nature. For this reason, maintaining a viable and healthy zoo population is challenging but urgently needed as a back-up for threatened cat species in the wild. Methods of assisted reproduction have gained significance to improve reproduction in captivity. These methods include, for example, egg collection and fertilization outside the body (IVF) and the transfer of embryos back to a recipient female. For the domestic cat, IVF is well-established, and also the transfer of embryos has already resulted in healthy offspring. In this study, the IVF system developed for the domestic cat was tested on the African lion. Sperm and eggs were collected from the gonads of lions that had been euthanized for age or population management reasons. The eggs were fertilized in a laboratory with fresh or frozen–thawed sperm. Embryos were generated, demonstrating that the domestic cat IVF system is also suitable for lions. Unlike in domestic cats, the lion samples had to be transported to the laboratory from zoos distributed all over Europe. The prolonged transportation led to a loss of quality of the eggs and sperm and contributed to lower overall fertilization success.

**Abstract:**

In the last 30–40 years, in vitro maturation (IVM) and fertilization (IVF) of domestic cat oocytes have been established as part of the panel of assisted reproduction technologies. As a representative of wild felids, the African lion is not yet considered endangered. Nevertheless, the zoo population management of the African lion itself as well as other closely related felids would benefit from the establishment of an IVF system. Here, we aimed to investigate the transferability of domestic cat IVF technology to the African lion. From the ovaries of 42 lionesses aged between 0.75 and 15 years, a total of 933 IVF-suitable oocytes were retrieved and subjected to IVM and IVF. The overall maturation rate was 40.6% and 18.9% of these oocytes cleaved after fertilization, respectively. Embryos were generated by intracytoplasmic sperm cell injection as well as co-culture with epididymal sperm. Improvements in the model system also led to an improved outcome with in vitro produced embryos in the lion. Compared to domestic cats, the transportation of gonads to a specialized laboratory was time-consuming and influenced oocyte quality negatively. In conclusion, the domestic cat IVF system is adoptable for the African lion, although success rates are still lower.

## 1. Introduction

In vitro fertilization (IVF) of domestic cat oocytes has been successfully established over the last 30–40 years [1,2,3] and is now the basis of research into related aspects of assisted reproduction such as in vitro embryo development or vitrification of feline oocytes and embryos [4,5,6]. Advances in the techniques have been made possible by the excellent availability of testes/epididymides and ovaries, which are, in many countries, consistently provided from animal welfare castration programs. Oocytes from these ovaries can be isolated but must undergo hormonally regulated in vitro maturation (IVM) before achieving the ability for fertilization. This is always necessary when using castration waste and has led to in vitro maturation in cats being considered an established method today with success rates of 50–70% [7,8]. On the male side, sperm can be collected from the epididymis after neutering as well. It can be used directly to fertilize oocytes or cryopreserved [9,10]. Fertilization of domestic cat oocytes is performed by co-culture of oocytes with sperm or by intracytoplasmic sperm injection, and in vitro produced embryos can be cultured up to the blastocyst stage [11]. Even though embryo transfer is not a standard method in the domestic cat, several litters of live kittens have resulted from this technique [5,12]. This established cascade of in vitro techniques makes the domestic cat an important model animal in assisted reproduction technologies (ARTs) for all other cat species [13,14].

One of the most common cat species kept in zoos is the African lion (*Panthera leo*). African lions are reproducing well in human care. Thus, there is a low demand to develop methods of assisted reproduction. However, few publications demonstrated successful semen collection and cryopreservation in African and Asiatic lions [15,16]. Laparoscopic oocyte retrieval and subsequent fertilization has been performed [17], and as part of the Felid-Gamete-Rescue-Project our research group was able to show that also in vitro maturation followed by intracytoplasmic sperm injection led to successful fertilization and embryonic development up to the blastocyst stage [18]. We were also able to cryopreserve lion oocytes for the first time [19]. The oocytes survived vitrification, matured in vitro, and cleaved subsequent to in vitro fertilization. 

Although the production of IVF embryos currently has little practical application for the captive African lion population, it is considered that the African lion might be an excellent model species to bridge the gap in ARTs between the domestic cat as an established model and different endangered species of the *Panthera* genus. such as the Asiatic lion (*Panthera leo persica*). Due to the high numbers of lions in zoos, the sample availability is much better than that for other *Panthera* species but is nevertheless limited due to life expectancy and rare castration events. 

The Felid Gamete Rescue Project has taken on the task of processing the gonads of deceased lions and all other cat species to explore and improve ARTs, including IVM, IVF, embryo culture, and cryopreservation techniques, and to preserve the existing genetic diversity of the populations in human care [20]. 

In this study, we would like to summarize our experience regarding in vitro maturation and fertilization of African lion oocytes. The samples were collected over a period of 15 years. Based on the sample origin, transport options, genetic value, number of isolated oocytes, quality of sperm, and the availability of homologous sperm, individual decisions were made as to how to process these samples to achieve the optimal output. Thus, this survey does not claim to be one long-term scientifically planned experiment. It is, rather, a unique experience report composed of many individual case studies. Despite some shortcomings, well-founded results can be derived and conclusions can be drawn as to where the cascade of IVF techniques in the African lion currently stands.

## 2. Materials and Methods

### 2.1. Chemicals and Reagents

All chemicals were purchased from Merck KGaA (Darmstadt, Germany) unless stated otherwise.

### 2.2. Animals

In the period from 2007 to 2021, gonads of African lionesses were obtained from several European zoos, safaris, and wildlife parks from animals that underwent euthanasia because of medical indication or population management reasons. The decision about the euthanasia, the implementation, and responsibility as well as the decision to donate the samples to our project were in the sole hand of the respective zoo. All animal procedures were in accordance with the local animal health regulations of the donating institutions. The animals were between 9 months and 15 years old (Table 1).

Animals donated on the same day were processed conjointly until 2015. * Lionesses originally had 18, 18, 55, and 28 oocytes, and one-half was vitrified immediately. All sperm samples were from the epididymis, unless stated otherwise. DC = domestic cat; UC = collected by urethral catheterization. A1, A2, B, C, D, E, F, and G media compositions listed in Table 2. In IVF experiments, maturation failure was only detectable after IVF and final fixation of oocytes; therefore, all utilized media were noted.

### 2.3. Transport of Ovaries until Processing

Ovaries were retrieved from the euthanized animals by the local vet immediately after death, washed with saline, and placed in sealed tubes with physiological saline solution. For transportation to the Felid IVF lab, they were kept cooled in a Styrofoam box with cool packs. Transport occurred in two different ways: ovaries were either stored and shipped within 28 h via a courier’s overnight express service or they were picked up in person, shifted to transport medium (T2; Table 2), and brought to the lab by car within 10 h.

### 2.4. Oocyte Retrieval and In Vitro Maturation (IVM)

Immediately upon arrival at the lab, ovaries were freed from surrounding tissues and washed in Washing Medium (WM; Table 2). In a few cases, ovaries were processed immediately on-site in a field laboratory temporarily set up in the zoo premises. 

To allow the release of the oocytes from the follicles, the ovaries were placed in Petri dishes containing WM and sliced with a scalpel. In the IVF lab, this was performed under the flow box. Good quality oocytes (with dark, homogeneous cytoplasm and several granulosa cell layers) were selected under a stereomicroscope for further processing. The oocytes were washed at least twice to remove attached blood cells and finally transferred to 4-well dishes containing maturation medium supplemented with human luteinizing hormone (LH) and human pituitary follicle-stimulating hormone (FSH). The specific medium composition has been changed over time and is presented in Table 2. 

In vitro maturation was performed in groups under embryo-tested mineral oil from different suppliers (Table 2) in a stationary or transportable incubator (CellTrans+, Labotect Labor-Technik-Göttingen GmbH, Rosdorf, Germany) at 38.5–39.0 °C in 5% CO_2_ in a humidified air atmosphere. Cumulus expansion was evaluated after 24 h of culture as one sign of maturation, since extrusion of the polar body was not visible due to several layers of surrounding cumulus cells. In the lion, cumulus expansion was not observable after 24 h; thus, IVM was extended in the initial experiments to up to 36 h. Removal of cumulus cells ahead of time (after 24 h) was not considered for two reasons. Cumulus cells are necessary to allow the completion of maturation of not yet fully matured oocytes and also to increase fertilization success rates via co-culture because initial sperm binding is mediated by the cumulus cells.

### 2.5. Fertilization of Oocytes

Depending on the available sperm sample and its quality, a decision was made to perform co-culture of oocytes and spermatozoa or intracytoplasmic sperm cell injection (ICSI). ICSI was chosen whenever sperm motility and/or concentration made fertilization success by co-culture improbable.

Sperm recovery, cryopreservation of sperm, thawing, and preparation for IVF and ICSI have been described before [20]. In brief, the cauda epididymis was separated from the testis and sliced with scissors in M199 (HEPES modification) to release the sperm. The sperm suspension was flushed through a 30-micrometer filter and centrifuged at 700× *g* for 8 min, and the pellet was resuspended in a small volume of sperm medium (Table 2). Cell concentration was determined with a counting chamber and the sample was adjusted to the desired concentration. For cryopreservation, the sperm suspension was diluted by two volumes of TestG buffer containing 7% (*v*/*v*) glycerol and cooled down in a refrigerator for 40 min. Cryopreservation occurred in the vapor of liquid nitrogen for 15 min to −167 °C. Finally, the vial was plunged in [9]. Samples were thawed by slewing a single vial for 90 s in a water bath at 38 °C. The sperm suspension was transferred to a pre-warmed 2-milliliter tube and diluted with an equal volume of pre-warmed sperm medium. Diluted spermatozoa were centrifuged (500× *g*, 5 min), resuspended in IVF medium, and used directly for ICSI or were adjusted to a final fertilization concentration of 1 × 10^6^ motile spermatozoa/mL for co-culture.

As presented in Table 1, co-culture of gametes was predominantly performed with frozen–thawed lion epididymal or, in one case, with cryopreserved urethral catheterized spermatozoa. Oocytes of three further lionesses were split. Half were subjected to ICSI with cryopreserved lion sperm, and the other half were co-cultured with fresh heterologous domestic cat epididymal sperm to evaluate fertilization potential. Oocytes were removed from the maturation dishes, washed twice, and transferred to 400 µL IVF medium (Table 2) covered by mineral oil in 4-well dishes. Co-incubation of oocytes and spermatozoa took place in the incubator for 18–24 h at 38.5–39 °C and 5% CO_2_ in a humidified air atmosphere. 

Intracytoplasmic sperm injection (ICSI) was performed according to [21] with small modifications. A 6-centimeter Petri dish (Nunc) was prepared with two 3–10-microliter droplets of a ready-to-use polyvinylpyrrolidone solution (PVP; Origio, Berlin, Germany or Gynemed, Lensahn, Germany). One of the drops was diluted 1:2 (*v*:*v*) in ICSI medium (WM with 3 mg/mL HEPES) to reduce viscosity and facilitate the movement of low-quality sperm. Further, nine 5-microliter droplets of ICSI medium were added and all drops were covered with mineral oil. Less than 1 µL of sperm solution was placed in each PVP drop and the oocytes were transferred to the remaining drops after being stripped of cumulus cells by gently pipetting with a micropipette (The Stripper, BioTipp, Waterford, Ireland). Each oocyte was assessed for morphology and extrusion of the first polar body as a sign of maturation and fertilization capability under an inverted microscope at 200× magnification (Axiovert 100; Carl Zeiss, Jena, Germany). Finally, a motile and morphologically normal spermatozoon was immobilized and injected into the mature oocytes from the 3 o’clock position after placing the polar body at the 6 or 12 o’clock position. Oocytes were fertilized using fresh or frozen–thawed epididymal lion sperm. In one case where fresh tiger sperm but only low-quality cryopreserved lion sperm were available, the decision was made to evaluate the developmental capacity of the 5 oocytes by fertilizing with heterologous fresh epididymal tiger spermatozoa.

### 2.6. Embryo Culture

Over the years, three different in vitro culture (IVC; Table 2) media were used and embryo culture was performed as single-cell [26] as well as group culture with up to 40 oocytes. Directly after microinjection by ICSI or after gamete co-incubation and subsequent washing and removal of residual cumulus cells and unbound sperm (Figure 1C), potential zygotes were transferred to 400 µL embryo culture medium, in 20-microliter microdrops in Petri dishes or in a time-lapse system (Primovision, Vitrolife, Västra Frölunda, Sweden) at 39 °C in 5% CO_2_ and 5% O_2_. All dishes were covered by mineral oil. Evaluation of embryo development was performed every 24 h for up to seven days (Figure 1D). The one-step culture medium was neither changed nor refreshed during embryo culture. In all experiments, non-cleaved oocytes and embryos arrested in their development were fixed in 96% ethanol overnight and then stained with propidium iodide (PI, 1.0 mg/mL, 1:100 in PBS; Thermo Fisher Scientific, Dreieich, Germany) to confirm their nuclear (maturation) status or developmental stage.

### 2.7. Vitrification

Oocytes and embryos were vitrified by the Cryotop method as previously described [19]. Briefly, one to eight oocytes were equilibrated at room temperature in an equilibration solution (ES) containing 7.5% (*v*/*v*) ethylene glycol (EG) and 7.5% dimethyl sulfoxide (Me_2_SO) in Medium 199 with 20% fetal bovine serum (FBS) for 15 min. Within the next 60–90 s, they were transferred into a vitrification solution (VS: 15% (*v*/*v*) EG, 15% Me_2_SO, and 0.5 M sucrose in Medium 199 with 20% FBS), placed on a Cryotop polypropylene strip, removed of the excess of liquid to reduce the volume as much as possible, and finally immersed into liquid nitrogen. 

### 2.8. Statistical Analysis

Comparisons of IVM/IVF rates were analyzed using two-tailed contingency tables and Fisher’s exact test. A Mann–Whitney U test was performed to compare the average age of lion groups; *p*-levels < 0.05 were considered significant in all tests. Statistical analysis was performed using R statistical software (R version 4.1.1 (10 August 2021)—“Kick Things”, R Foundation for Statistical Computing, Vienna, Austria).

## 3. Results

In this survey, data from 42 lionesses were included and a total of 933 oocytes with in vitro culture suitability were obtained (Table 1). The mean number of retrieved oocytes per animal was 22.2, ranging between 2 and 72 oocytes per individual. No oocytes suitable for IVM/IVF in vitro culture were found in five additional animals. These animals, aged 5 days, 10 weeks (2 animals), and 7 and 17 years, were therefore not included in this survey. Immediately after isolation, 874 lion oocytes were subjected to culture, whereas 59 good quality oocytes were frozen by vitrification (see specific results in [19]). After 24–36 h of IVM, 355 oocytes or 40.6% were classified as mature either by determining the nuclear stage with PI or the visibility of extruded polar bodies or by cleavage after fertilization. Oocytes of one adult and one prepubertal lioness were subjected to IVM only and were therefore excluded for calculation of cleavage. Finally, 64 embryos were produced (18.9%). It was shown that occasionally, embryo development could proceed to the blastocyst stage, although the rate was still insufficient. To avoid prolonged culture in vitro, good quality embryos of early stages were frozen for biobanking. Currently, twenty-one embryos of all stages have been vitrified and stored. For these embryos, the developmental potency is unknown, making overall calculation of developmental rates impossible.

### 3.1. Impact of Age of Animals on Oocyte Numbers and IVM/IVF Potential

Independently from the variations of the methods used, it is well known from humans that age has a dramatic impact on reproduction parameters such as oocyte number, quality, and fertilization capacity [27]. 

In this data collection, 17 animals were prepubertal, with an age of less than 2 years. The remaining 25 animals were adults in the age range of 5 to 15 years (Table 1). The average number of oocytes obtained per juvenile (prepubertal) animal was 24.6 oocytes/lioness, slightly higher compared to 20.6 oocytes per adult animal (Table 3). Although the maturation rate of oocytes did not differ between the age groups (42.6% vs. 38.8%), the fertilization rate was significantly higher in the prepubertals (24.5% cleavage) vs. adults (13.7% cleaved oocytes; *p* = 0.0127).

### 3.2. Method Changes over Time

Overall, we observed an increase in our IVF/IVM outcome over time, which surely contributed to the permanent advancement of our methods. When comparing maturation and fertilization rates from animals processed between 2007 and 2014 with animals processed between 2015 and 2021, the maturation rate increased significantly (*p* = 0.0001) from 28.2% in the early phase to 51.3% in the latter period of the project. In contrast, the cleavage rate of the two groups did not differ over the course of the years (15.5% for the early period vs. 20.3% for the latter period). There was no difference in the mean age of lionesses for both groups.

In the following, we relate the different parameter changes (transportation conditions, medium composition, maturation duration, fertilization method) over the time course of the project to the maturation and cleavage outcomes for lion oocytes to unravel important (relevant) factors which contributed to the observed improvement (Table 1 and Table 2).

#### 3.2.1. Method Variations without Impact on IVM/IVF Outcome

Depending on the availability and quality of sperm, co-culture or ICSI was chosen as the fertilization method. In three cases, a direct comparison between both fertilization methods was performed with cryopreserved epididymal lion sperm for ICSI and fresh epididymal domestic cat sperm in co-culture, because the lion sperm quality was not suitable for IVF. Most other IVF co-cultures were conducted with cryopreserved lion epididymal sperm (*n* = 9) but one was conducted with cryopreserved lion urethral catheterized sperm [15] and one with fresh epididymal lion sperm. In case of ICSI (*n* = 26 animals), fertilization was carried out once with fresh epididymal tiger sperm and once with fresh epididymal lion sperm. All other ICSI procedures were conducted with cryopreserved epididymal lion sperm. Overall, there was no difference in fertilization success between co-culture IVF and ICSI (20.2% vs. 17.4%; Table 4), although the number of obtained blastocysts seems to be higher for the ICSI group (4 of 27 embryos) compared to the co-culture group (1 of 37 embryos). Due to the low embryo number, a statistical analysis was not possible.

In addition to the fertilization method, the culture conditions for maturation, fertilization, and embryo development have been varied over time. For instance, embryos were kept singly in small drops of 20 µL or in groups with up to 40 oocytes in drops of 400 µL media. The single culture during the embryo culture was introduced after approving in the domestic cat system to track the individual developmental fate of all oocytes [19,26]. There were no indications (data not shown) that the chosen culture system had an impact on the IVM/IVF outcomes, although due to the limited number of trials, a real statistical evaluation was not (yet) possible. Variation of the culture method was also performed in regard to media compositions (Table 2), but all were introduced according to changes validated in the domestic cat.

#### 3.2.2. Method Variations with Impact on IVM/IVF Outcome

One important factor which seemed to influence IVM outcome was found to be the time frame between the death of the animal and oocyte retrieval from the ovary. The best maturation and fertilization results were obtained when the samples were processed immediately on-site in the premises of the zoo (IVM: 50.1%, IVF: 23.2%; Table 5). In these cases, oocytes were matured in a portable incubator. However, on-site processing occasions were accompanied by an extensive change of the IVM culture medium from M199-based medium to commercially available Quinn’s Advantage Protein Plus™ Blastocyst Medium. This change would not have been made if experience with the domestic cat had not indicated a clear benefit for the maturation rate (data not shown). Since the reduction of the transportation time and the medium change took place in parallel, it is not possible to distinguish which factor contributed to the positive effect observed on oocyte maturation in lion and to what extent. Statistical validation had therefore to be dispensed with.

If the gonads needed to be transported (cooled in Styrofoam boxes), the maturation rates dropped to 38% for 10 h shipment and 25.6% for 24–28 h shipment. A similar trend was observed also in the number of cleaved embryos. 

### 3.3. Duration of Maturation

The duration of maturation also influenced the outcome of IVM/IVF significantly (Table 6). In contrast to the domestic cat [28], initially the maturation was extended. This was based on the observation that expansion of the cumulus cells could hardly ever be found after 24 h of IVM compared to domestic cats (Figure 1A,B). Therefore, the maturation time for a total of 544 oocytes from 28 lionesses was set to 32–36 h. Even after that, the cumulus expansion was low. The rate of maturation for these oocytes was 38.8% (211/544 oocytes), followed by a cleavage rate of 12.8%. For technical reasons (fertilization after 32–36 h takes place in the middle of the night), a shortening of the maturation time was considered. Thus, 235 oocytes of 13 different lionesses were matured only for 24–28 h, resulting in significantly (*p* = 0.0091) higher maturation (115/235 oocytes; 48.9%) and fertilization rates (25.2%; *p* = 0.0057). In some cases, oocytes of single lionesses were divided into both maturation groups, making the result even more reliable. Ninety-five oocytes of an intermediate maturation time between 29 and 31 h were excluded from this analysis.

## 4. Discussion

In this study, IVF results from 42 lionesses collected in zoos from 2007 to 2021 are presented for the first time. As a summary of many individual and sometimes sudden occasions, it could not be designed in advance as one carefully controlled long-term experiment. This means that neither the time of sample availability and the number of individuals nor their age could be controlled or chosen. Sudden cases of euthanasia due to health issues impaired the careful planning of transportation and the individual experiment. Depending on the received quality of gonads, changes in processing and the cultivation process needed to be applied spontaneously. Nevertheless, compiling all data to extract the factors responsible for the progress that has been made during the long-term project might be indicative for the future planning of gamete rescue projects in wildlife species. 

Random factors such as animal health and age not only complicated the experimental design but also influenced oocyte quality and developmental potential. Slightly more oocytes could be retrieved from prepubertal females, and surprisingly, developmental competence was also higher than after puberty. This is in contrast to the findings of other researchers who described superior developmental potential of oocytes of adult animals in mice, sheep, cattle, and felids [29,30,31,32], but this might be founded in the heterogeneity of the adult lioness group, including severely sick, hormonally affected (e.g., due to contraception), or post-reproductive females. Post-reproduction under human care conditions means the absence of pregnancies in formerly proven breeders and can be overcome to some extent by assisted reproduction as long as oocyte retrieval is still possible. This is well-known, especially in humans [33]. However, the fertilization probability of individual oocytes is reduced by aging processes such as chromosomal aneuploidy or mitochondrial dysfunction [34]. For a gamete rescue project, this means that even in aged animals, oocyte collection from the ovaries for biobanking makes sense and is the last chance, even though the developmental potential of individual oocytes is reduced. These facts are mirrored in two examples of this survey: Three of five oocytes of one 13-year-old female were fertilized and one developed to the morula stage. On the other hand, no oocytes could be found in one 17-year-old female. Thus, zoos should consider gonad collection before age-related euthanasia is required. Longevity of animals in zoos is not necessarily correlated with prolonged fertility, and the number and quality of oocytes drop down to zero in very old (post-menopausal) individuals, making an oocyte rescue impossible [20]. 

In addition, a timely planned euthanasia or castration procedure for either population management or animals with health issues allows for sophisticated preparation of oocytes collection and planning of experiments. For example, with planned events we were able to compare different media compositions directly [18]. A planned gonad collection would also allow to optimize the transportation of gonads to a specialized laboratory, or even to organize on-site gamete processing, which was possible in this survey only because the respective animals were euthanized for population management and not for health reasons. The data presented here give a clear indication that, in addition to a suitable culture system, the initial quality of the sample material has a key significance for IVM/IVF success and that this is decisively influenced by the required transport time (Table 5). Cooled transportation of oocytes within the ovaries for up to 24 h to a specialized laboratory is possible and common practice in genome resource banking of wildlife animals [20,35,36,37]. However, this delayed gamete retrieval induces a certain quality loss due to atresia [24,38,39,40] in addition to the uncalculated risks of temperature fluctuations, bacterial contamination, and further unforeseen delays caused by crossing country borders. At least for the African lion, the reduction in transportation time to less than 10 h achieved by direct pick-up of the gonads from the zoos already remarkably increased the maturation and cleavage rates, whereas direct processing on-site and immediate maturation of oocytes in a transportable incubator seem to be the required option if the donating female is of high genetic importance. On-site collection would also allow immediate cryopreservation of oocytes, as we already demonstrated for African lion [19]. Successful oocyte cryopreservation offers greater flexibility in terms of time and place, both during transport and at the start of the culture, and independent of immediate availability of the matching counterpart. Cryopreservation of lion gametes was not the focus of this study but has been performed successfully for both sperm and oocytes [15,19,41]. As with domestic cats, however, the resulting embryos of cryopreserved oocytes have a reduced development potential [4,42,43,44], and further research is needed to improve freezing techniques.

Our compiled results also allowed us to determine the required time for lion oocytes to mature in vitro. The maturation of feline oocytes is a well-synchronized process comprising nuclear and cytoplasmic maturation and expansion and loosening of surrounding cumulus cells. In the domestic cat, the most studied feline species, in vitro maturation is known to require about 24–28 h [3,8,25,28,45,46]. Due to the high lipid content within feline oocytes and the several layers of cumulus cells around the oocyte, cumulus expansion is the only visible hint for maturation. Complete removal of the cumulus e.g., for ICSI visualizes the extruded polar body as a clear indication of nuclear maturation. But the removal of the cumulus also interrupts interactions between cumulus cells and the oocyte and hampers completion of still onging maturation processes. With planned IVF by co-culture removal of cumulus cells reduces subsequent fertilization success. Sperm binding as the first step of fertilization procedure is mediated by necessary interaction of enzymes on the sperm head surface with the cumulus matrix [47,48,49].

In contrast to domestic cats, however, extensive cumulus expansion in lion oocytes could practically never be observed, and maturation was therefore previously extended to 32–36 h. The optimal insemination time point was defined as after 32 h for the domestic cat [3], but with progress in culture systems, contemporary research groups tend to reduce maturation to 24 h [26,29]. However, a domestic-cat-adapted culture system might be suboptimal for African lions, suggesting a prolonged time is necessary to complete maturation. 

Even after 36 h of IVM, cumulus expansion was still low and the maturation rate of 38.8% was lower than that in the domestic cat. Poorer maturation rates can also be explained by a less strict selection of best quality oocytes in rare wildlife animal samples. Since cumulus expansion under in vitro conditions in African lions is obviously not a good maturation marker, the decision was made to verify maturation after 24–28 h. Surprisingly, the maturation rate of these oocytes was significantly higher (48.9%) than that under prolonged maturation conditions. These findings can be explained by an aging process of oocytes including spindle degeneration after achieved maturation [50,51]. However, the proportion of oocytes that degenerated as a result of prolonged maturation cannot be precisely determined in retrospect because the oocytes were often cultivated for too long in order not to accidentally fix oocytes with the potential to divide. 

Compared to domestic cats, the overall fertilization rates of roughly 17–20% expressed at first cell division were not satisfactory, with either IVF or ICSI. The fact that fertilization success with ICSI was not superior to co-culture underlines that the reasons are to be sought on the female side and might be associated with impaired oocyte quality due to less strong selection, invisible transportation damage, suboptimal culture conditions, or incomplete or defective maturation of the oocyte. In this survey, the fertilization rate was calculated as the proportion of matured oocytes and should therefore not differ between short and prolonged maturation. Surprisingly, fertilization after prolonged maturation led to a significantly lower division rate than oocytes that matured for 24–28 h (12.8% vs. 25.2%). The most probable explanation is that the optimal timeline for fertilization had passed and the oocytes already reached an early but invisible degeneration stage [50,51]. Finally, the proportion of parthenogenetic divisions was not examined here due to the rarity of samples but should be taken into account in future studies [52]. 

During this survey, the rate of maturation has improved significantly from 28.2% (2007 to 2014) to 51.3% (2015 to 2021). A multitude of small adaptations during the years were inevitable and could have led to the enhancement in summary. Changes in hormone supplements and concentrations were always tested in the domestic cat system first before being adopted for the lions. This also applies to the change in the culture medium from Medium 199 with various additives to the commercially available Quinn’s Advantage Protein Plus™ Blastocyst Medium. However, the increase in maturation rates to 70–80% in domestic cats (data not published) has not been proven in lions yet. 

## 5. Summary/Conclusions

The present survey demonstrates that the domestic cat IVM/IVF system is a good starting point to establish IVF techniques in African lions, though the success rates for maturation and also fertilization with co-culture or ICSI are still lower. Cooled transportation of fresh gonads is possible as well as on-site processing under extemporized laboratory conditions. In the case of very valuable females, on-site oocyte retrieval is highly preferable to minimize transportation risks and quality loss between gonad collection and processing. Oocytes of prepubertal females as well as aged adults are worth cultivating, and embryonic development can reach the blastocyst stage.

The process of developing and optimizing the efficiency of the IVF cascade in African lions is obviously not yet complete, and further techniques such as cryopreservation of oocytes and embryos as well as embryo transfer need to be tackled.

## Figures and Tables

**Figure 1 animals-12-01424-f001:**
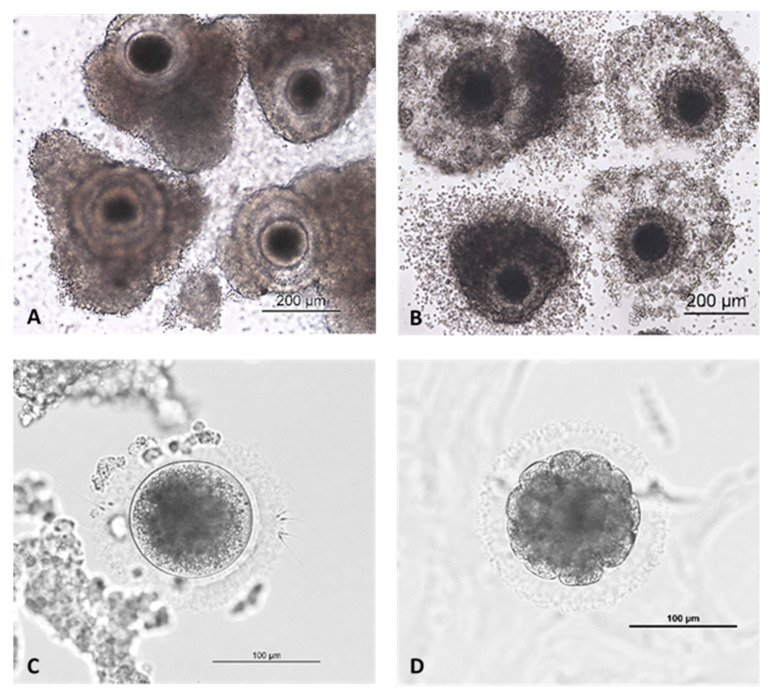
(**A**) Low cumulus expansion of African lion oocytes after 24 h in vitro maturation. (**B**) Cumulus expansion of domestic cat oocytes after 24 h in vitro maturation. (**C**) Sperm bound to zona pellucida after co-culture (African lion). (**D**) Early African lion morula, day 4.

**Table 1 animals-12-01424-t001:** Parameters and outcomes of IVF cascade for all lionesses.

African Lion	Year	Age (years)	Transport (h)	n Oocytes	Duration of IVM (h)	n Matured Oocytes	Maturation Rate (%)	n Embryos	Ferilization Rate (%)	ICSI/IVF	Sperm Source	Utilized Media
1	2007	5	28	22	36	6	27.3	0	0	IVF	lion cryo	A1, C, E
2	2008	15	24	31	36	2	6.5	−	−	−	−	A1 (IVM only)
3	2008	1	24	31	36	15	48.4	−	−	−	−	A1 (IVM only)
4	2009	13	5	4	36	0	0	−	−	−	−	A1, C, E
5	2009	7	10		36							A1, E
6		7	10	71		19	26.8	0	0	ICSI	lion cryo	
7		7	10									
8	2010	1	6	40	34	8	20	1	12.5	ICSI	lion cryo	A1, E
9		1	6									
10	2010	5	24	26	28	5	19.2	2	40	ICSI	tiger fresh	A1, E
11	2011	8	10		35				0			A1, C, E
12		7	10	37		18	48.6	0		IVF	lion cryo	
13		6	10									
14	2012	1	24	4	24	1	25	0	0	ICSI	lion cryo	A1, E
15	2012	1	24	3	24	0	0	−	−	−	−	A1
16	2012	14	10		36				9.1			
17		6	10	55		11	20	1		ICSI	lion cryo	A1, E
18		6	10									
19	2013	1	10	70	34	25	35.7	11	44	ICSI	lion cryo	A1, A2, F
20		1	10									
21		1	10									
22		1	10									
23	2014	0.75	24	10	28	4	40	0	0	ICSI	lion cryo	A1, F
24	2015	1	10	66	35	49					lion cryo/ DC fresh	A1, C, F
25		1	10				74.2	6	12.2	ICSI/IVF		
26		1	10									
27	2016	14	10	14	31	6	42.9	2	33.3	IVF	lion cryo UC	A1, D, F
28	2017	7	8	4	30	1	25	0	0	ICSI	lion cryo	A1, F
29	2018	11	28	23	25	5	21.7	0	0	IVF	lion cryo	A1, D, F
30	2019	7	24	8	24	3	37.5	0	0	ICSI	lion cryo	A2, F
31	2019	6	20	2	28	0	0	−	−	−	−	A1
32	2019	7	0	9 *	32−34	5	55.6	0	0	ICSI	lion cryo	B, F
33	2019	7	0	9 *	32−34	1	11.1	0	0	ICSI	lion cryo	B, F
34	2019	5	0	28 *	32−34	17	60.7	1	5.9	ICSI	lion cryo	B, F
35	2019	5	0	14 *	32−34	10	71.4	2	20	ICSI	lion cryo	B, F
36	2020	1	0	54	30	8	44	0	0	ICSI	lion fresh	B, F
37	2021	13	0	10	28	5	50.0	3	60	IVF	lion cryo	B, D, F
38	2021	9	0	72	28	48	66.7	10	20.8	IVF	lion cryo/fresh	B, D, F
39	2021	9	0	17	28	15	88.2	3	20	IVF	lion fresh	B, D, F
40	2021	1	0	47	26/32	23	48.9	5	21.7	IVF	lion cryo	B, D, G
41	2021	1	0	23	29	14	60.9	6	42.9	ICSI	lion cryo	B, G
42	2021	1	0	70	26/32	31	44.3	11	35.5	IVF	lion cryo	B, D, G
Total		Ø 5.3		874 (933)		355	40.6	64	18.9			

A, B, C…: different culture media in alphabetical order; A1, A2: Identical basic medium but different concentration of supplements.

**Table 2 animals-12-01424-t002:** Overview of media compositions utilized in this survey.

Event	Basic Medium	Supplements	References
Transport:			
T1	Physiological saline solution	-	
T2	Hepes-MEM (M7278)	3 mg/mL BSA, 1:100 (*v*:*v*) Antibiotic Antimycotic Solution	[21]
Washing medium:			
WM	M199 (M4530)	3 mg/mL BSA, 1.4 mg/mL HEPES, 0.6 mg/mL sodium lactate, 0.25 mg/mL sodium pyruvate, 0.15 mg/mL L-glutamine, 0.1 mg/mL cysteine and 0.055 mg/mL gentamicin	[21]
IVM:			
A1	M199 (M4530)	WM + 0.05 IU/ml luteinizing hormone (LH) + 0.02 IU/ml pituitary follicle-stimulating hormone (FSH)	[21]
A2	M199 (M4530)	WM + 10x LH and FSH	[22]
B	Quinn’s Advantage Protein Plus™ Blastocyst Medium (ART-1529)	+ 10x LH and FSH	[19]
IVF:			
C	Tyrode’s salts solution (T2397)	6 mg/mL BSA, 1.2 mg/mL HEPES, 1.1 mg/mL sodium lactate, 0.1 mg/mL sodium pyruvate, 0.15 mg/mL L-glutamine, 2.2 IU/mL Heparin	[23]
D	M199 (M4530)	WM + 2.2 IU/mL Heparin	[24]
IVC:			
E	M16	3 mg/mL BSA, 0.03 mg/ml gentamicin, 0.1 mM non-essential amino acids	[21]
F	HAM’s F10 (N6013)	5% FBS, 0.11 mg/mL sodium pyruvate, 0.15 mg/mL L-glutamine, 0.06 mg/ml gentamicin	[22]
G	Quinn’s Advantage Protein Plus™ Blastocyst Medium (ART-1529)	5% FBS	[25]
Sperm media:			
H	Tyrode’s salts solution (T2397)	6 mg/mL BSA, 1.2 mg/mL HEPES, 1.1 mg/mL sodium lactate, 0.1 mg/mL sodium pyruvate, 0.15 mg/mL L-glutamine	[21]
I	M199 (M7528)		[9]
Mineral oil:			
	Sigma-Aldrich (M8410)		[21]
	Reprodline Medical GmbH (REF451200)		[24]
	Cooper Surgical (ART-4008-5)		not published

T for Transport; WM for Washing medium; A, B, C…: different culture media in alphabetical order; A1, A2: Identical basic medium but different concentration of supplements.

**Table 3 animals-12-01424-t003:** Outcome of IVM and IVF from lion oocytes obtained from prepubertal (<2 years) and adult lionesses.

Lionesses	*n*	No. of Oocytes	No. of Oocytes/Animal	Maturation Rate	Cleavage Rate
Prepubertals	17	418	24.6	42.6% ^a^ (178 of 418)	24.5% ^a^ (40 of 163)
Adults	25	515	20.6	38.8% ^a^ (177 of 456)	13.7% ^b^ (24 of 175)
Total	42	933	22.2	40.6% (355 of 874)	18.9% (64 of 355)

Maturation rate and cleavage rate are presented as percentages based on the number of oocytes used in the respective experiments (in brackets). Different superscripts (a, b) within the same column indicate significant differences among groups (*p* < 0.05).

**Table 4 animals-12-01424-t004:** Cleavage and embryo development as outcomes of fertilization method. Due to the low numbers, statistical analysis of embryo development was omitted.

Fertilization	*n* Lionesses	No. of Matured Oocytes	Cleavage Rate	Stage of Embryo		
				2–16 Cells	Morula	Blastocyst
Co-Culture	14	183	20.2% ^a^ (37 of 183)	33 [12]	3 [3]	1 [0]
ICSI	26	155	17.4% ^a^ (27 of 155)	22 [5]	1 [0]	4 [1]
Total	40	338	18.9% (64 of 338)	55	4	5

Cleavage rates are presented as percentages based on the number of oocytes used in the respective experiments (in brackets). Numbers of vitrified embryos of the different stages are depicted in square brackets. Due to the low numbers, statistical analysis of embryo development was omitted. Different superscripts (a, b) within the same column indicate significant differences among groups (*p* < 0.05).

**Table 5 animals-12-01424-t005:** Impact of time between retrieval of gonads and oocytes collection. For transportation, ovaries were stored in physiological salt solution in a Styrofoam box equipped with cool packs.

Transport Duration (Location of Oocyte Collection)	*n* Lionesses	Mean Age of Lionesses	No. of Oocytes	Maturation Rate	Cleavage Rate
0 h (on-site collection)	11	5.3	412	50.1% (177 of 353)	23.2% (41 of 177)
10 h (IVF lab)	21	5.3	361	38.0% (137 of 36)	15.3% (21 of 137)
24–28 h (IVF lab)	10	5.3	160	25.6% (41 of 160)	8.3% (2 of 24)

Maturation rates and cleavage rates are presented as percentages based on the number of oocytes used in the respective experiments (in brackets).

**Table 6 animals-12-01424-t006:** Effects of the duration of IVM in African lion oocytes.

Maturation Time	*n* Lionesses	No. of Oocytes	Matured Oocytes *n* (%)	Cleaved Embryos *n* (%)
24–28 h	13	235	115 ^a^ (48.9)	29 ^a^ (25.2)
32–36 h	28	544	211 ^b^ (38.8)	27 ^b^ (12.8)

Different superscripts (a, b) within the same column indicate significant differences among groups (*p* < 0.05).

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
