# Peer review of "Current State of In Vitro Embryo Production in African Lion (Panthera leo)"

_animals, 2022, doi:10.3390/ani12111424_

Round 1
Reviewer 1 Report
The authors applied the in-vitro fertilization method on wild felines that was used mostly for domestic cats. I think the authors have done a decent job after considering several parameters for selecting the subjects and performing the experiments. And I am completely aware that this is not an easy task to collect ovum and sperm from wild felines. However, I would still suggest that authors should have tried to improve the overall rate of successful implantation. I recommend considering the manuscript in its current form.
Reviewer 2 Report
Dear Authors
The paper "Current State of in-vitro fertilization in an African lion (Panthera leo)" submitted for review meets the requirements of Animals to be published in.
I want to underline that the presented research carry scientific and practical value. The following remarks need to be addressed before consideration for publication.
The introduction to the work clearly explains the purpose of the topic. The paper presents an important contribution to the field of clinical embryology and protective biological material from valuable wild Felids.
The part "Material and methods" explained all stages of the experiment and applied research methods in detail. Due to the extensive methodology: changes in culture media, duration of transport and maturation, and fertilization technique, the addition tables clearly track the research methodology.
Some questions are included below:
Line 125: Why IVM was extended to up to 36h? Nuclear maturation is more important, so what was the point of extending maturation? Please add stronger arguments for this decision in the Introduction part.
Line 149 and Line 170: Do the Autors have the consent of the ethics committee (add consent number) for interspecies IVF fertilization in the case of endangered animals ( used domestic cat and tiger spermatozoa semen to IVF).
Line 191: "...The medium was not changed during embryo culture…" If it was the one-step medium, please add if the medium was refreshed.
Line 219 and 227: In Results, there is information about vitrification oocytes and embryos, but was missed in Materials and Methods. Please add information to the Materials and Methods part about methods, devices, and media composition to vitrification.
Results obtained in the study are well presented and compared with the literature. In part Conclusion, there is no clear summary about the results obtained in the study: the time of transport material, collected oocytes or the IVM time. There is only a general summary. Please add a synthetic conclusion about obtained results.
In my opinion, The paper "Current State of in-vitro fertilization in an African lion (Panthera leo)" after revision consideration for publication in Animals.
Reviewer 3 Report
Animals -1658419- Current State of in-vitro fertilization in African lion (Panthera leo)
General Comments
This manuscript is a retrospective analysis describing individual studies on in vitro maturation, in vitro fertilization, and ICSI of African lioness oocytes during the last 15 years. The topic is engaging since reproductive technologies are necessary for gamete rescue projects in wildlife species. However, this manuscript is written in a messy way that is difficult to follow. There are many inaccuracies in the description of the methodologies, and many comparisons are not correct.
It is not clear why this manuscript is written in the format of an Original Research instead of a Review, analyzing the different factors involved in these reproductive technologies and how these factors affect the outcomes.
Specific Comments
The manuscript title should change since it refers only to in vitro fertilization but the manuscript topic involves ICSI, in vitro maturation, embryo development, and cryopreservation; therefore, reproductive biotechnologies would be more appropriate than in vitro fertilization.
The reasons for euthanasia should be better explained, as well as the procedures used to obtain the samples.
The material and methods section is poorly described, not well written, and described as if it were an individual study and not different studies. In addition, there are misconceptions, such as the first polocyte as a sign of fertilization instead of the second polocyte after fertilization.
This manuscript should be rewritten as a review with a better analysis of the factors involved in reproductive technologies of wild cats, with an emphasis on in vitro fertilization.
Round 2
Reviewer 3 Report
The revised version of this manuscript addressed some suggestions, but unfortunately, these changes have not improved the paper. The main criticism given in the first review was not considered. Why do the authors write this manuscript as an original study rather than a review? At the end of the Introduction section, they say, “In this study, we would like to summarise our experience regarding in-vitro maturation and fertilisation of African lion oocyte “. This is a review. A research paper has specific methods and a clear objective. Here, the maturation protocols, sperm handling, cryopreservation, fertilization, etc., are mixed, which does not correspond to one investigation but to several research methodologies and objectives. Those results should be analyzed in the context of a review.
On the other hand, it is very confusing to the analysis of results over 15 years to perform statistical analysis. What is statistically compared when there are different methodologies and different protocols? In addition, the authors point out a null hypothesis (“ In order to test the null hypothesis that the transportation time does not influence maturation or cleavage rate, a Pearson’s chi-squared test was conducted. “) that does not make any sense in the context of different studies. In addition, the conclusion or conclusions have nothing to do with this hypothesis, which seems to be entirely out of context.
In addition, there is no acceptable explanation for euthanasia, especially in young animals. To say that the reasons depend on each Zoo is not acceptable in ethical terms for being published.
I do not think that this manuscript presented in this form is suitable for publication.
Author Response
Dear Reviewer 3,
We greatly appreciate your extensive efforts to improve our manuscript!
1) In accordance with your suggestion we adapted our manuscript and are submitting the article now as a review.
2) After consultation with our statistician, we agree that the statistical evaluation of the transport duration is not valid. All on-site processing occasions are accompanied by a change of the base maturation medium and the influence of both factors cannot be separated from each other. For this reason, we do not perform a statistical evaluation of transportation conditions.
3) Regarding the euthanasia of young animals we added the following information to the Institutional Review Board Statement:
The utilization of oocytes for IVF experiments from domestic cat ovaries obtained by ovariectomy was approved by the Internal Committee for Ethics and Animal Welfare of the Leibniz Institute for Zoo and Wildlife Research in Berlin, Germany (Permit numbers: 2010-10-01).
The African lions whose samples were used in this study were not euthanized or treated in any way for the purpose of this survey. The decision as to whether an animal was euthanized for population management or veterinary reasons was the sole responsibility of the zoo in question and was made in accordance with local animal health regulations. All zoos that practiced culling of individuals for population management reasons in this survey are members of the European Association of Zoos and Aquaria (EAZA) and acted in accordance with the EAZA Culling Statement (https://www.eaza.net/assets/Uploads/Position-statements/EAZA-Culling-statement.pdf).
Kind regards,
Jennifer Zahmel

Round 3
Reviewer 3 Report
The manuscript has improved by describing the studies as a review, however the authors must change the structure of the manuscript in the context of a review.